# The Use of Levosimendan after Out-of-Hospital Cardiac Arrest and Its Association with Outcome—An Observational Study

**DOI:** 10.3390/jcm11092621

**Published:** 2022-05-06

**Authors:** Susanne Rysz, Malin Jonsson Fagerlund, Johan Lundberg, Mattias Ringh, Jacob Hollenberg, Marcus Lindgren, Martin Jonsson, Therese Djärv, Per Nordberg

**Affiliations:** 1Function Perioperative Medicine and Intensive Care, Karolinska University Hospital, 171 76 Stockholm, Sweden; susanne.rysz@regionstockholm.se (S.R.); malin.jonsson-fagerlund@regionstockholm.se (M.J.F.); 2Department of Medicine Solna, Karolinska Institutet, 171 77 Stockholm, Sweden; therese.djarv@ki.se; 3Department of Physiology and Pharmacology, Karolinska Institutet, 171 77 Stockholm, Sweden; 4Department of Clinical Neuroscience, Karolinska Institutet, 171 77 Stockholm, Sweden; johan.lundberg@regionstockholm.se; 5Department of Neuroradiology, Karolinska University Hospital, 171 76 Stockholm, Sweden; 6Center for Resuscitation Science, Department of Clinical Science and Education, Karolinska Institutet Södersjukhuset, 118 83 Stockholm, Sweden; mattias.ringh@ki.se (M.R.); jacob.hollenberg@ki.se (J.H.); martin.k.jonsson@regionstockholm.se (M.J.); 7Department of Medicine, Piteå Hospital, 941 50 Piteå, Sweden; marcus@tegenaria.com; 8Function Emergency Medicine, Karolinska University Hospital, 171 76 Stockholm, Sweden

**Keywords:** cardiac arrest, intensive care, levosimendan, inotropy

## Abstract

Background: Levosimendan improves resuscitation rates and cardiac performance in animal cardiac arrest models. The aim of this study was to describe the use of levosimendan in out-of-hospital cardiac arrest (OHCA) patients and its association with outcome. Methods: A retrospective observational study of OHCA patients admitted to six intensive care units in Stockholm, Sweden, between 2010 and 2016. Patients treated with levosimendan within 24 h from admission were compared with those not treated with levosimendan. Propensity score matching and multivariable logistic regression analysis were used to assess the association between levosimendan treatment and 30-day mortality Results: Levosimendan treatment was initiated in 94/940 (10%) patients within 24 h. The proportion of men (81%, vs. 67%, *p* = 0.007), initial shockable rhythm (66% vs. 37%, *p* < 0.001), acute myocardial infarction, AMI (47% vs. 24%, *p* < 0.001) and need for vasoactive support (98% vs. 61%, *p* < 0.001) were higher among patients treated with levosimendan. After adjustment for age, sex, bystander cardiopulmonary resuscitation, witnessed status, initial rhythm and AMI, the odds ratio (OR) for 30-day mortality in the levosimendan group compared to the no-levosimendan group was 0.94 (95% Confidence interval [CI], 0.56–1.57, *p* = 0.82). Similar results were seen when using a propensity score analysis comparing patients with circulatory shock. Conclusions: In this observational study of OHCA patients, levosimendan was used in a limited patient group, most often in those with initial shockable rhythms, acute myocardial infarction and with a high need for vasopressors. In this limited patient cohort, levosimendan treatment was not associated with 30-day mortality. However, a better matching of patient factors and indications for use is required to derive conclusions on associations with outcome.

## 1. Introduction

Out-of-hospital cardiac arrest (OHCA) affects more than 300,000 people annually in Europe, with a mortality rate of approximately 90% [1]. However, the systematic and broad introduction of basic and advanced cardiopulmonary life support in combination with a more standardised form of post-resuscitation care over the last decades has improved its prognosis [2,3,4]. To further increase survival and improved cardiac and neurologic function following OHCA, an exploration of pharmacological interventions beyond adrenaline and amiodarone is warranted [5,6].

Levosimendan is an inotropic substance used in acute severe heart failure (HF). Besides the inotropic effect due to calcium sensitization, levosimendan’s pharmacological mechanisms also include vessel wall smooth muscle relaxation, and cell protection by activating ATP-sensitive potassium channels (K_ATP_-channels) [7]. The lusitropic effect of levosimendan with reduced ventricular filling pressure and improved cardiac performance could add further potential advantages following OHCA, especially when considering that stunning and cardiogenic shock are common findings together with both systolic and diastolic dysfunction in cardiac-arrest patients [8,9,10,11]. Thus, levosimendan is one of several nonadrenergic vasoactive and inotropic agents that may be used to limit the reduced response to adrenergic agonists after prolonged stimulation, such as in cases of refractory circulatory shock [12].

Using an animal model, we have shown that these pharmacological effects might be beneficial when levosimendan is administered during resuscitation and in the post-resuscitation period following cardiac arrest [13]. Experimental studies with levosimendan in both ischemic and non-ischemic cardiac arrest models have shown promising results in terms of resuscitation rates, hemodynamic performance and survival [14,15,16,17]. However, in clinical practice, the use of levosimendan has not been studied properly in the context of circulatory shock following a cardiac arrest. There may be several reasons for this, including its vasodilating properties in patients in severe shock and its long half-life time. This may explain the limited data available on levosimendan use in OHCA patients [18]. The aim of this observational study was to describe the use of levosimendan in out-of-hospital cardiac arrest (OHCA) patients and its association with outcome.

## 2. Methods

### 2.1. Study Design and Ethics

We performed a retrospective observational study by extracting data from three different national registers (Swedish Registry for Cardiopulmonary Resuscitation (SRCR), Centricity critical care (CCC), and SWEDEHEART). The study was approved by Stockholm regional ethics committee (id: 2016/873-31/2) and was conducted according to the Helsinki declaration.

### 2.2. Study Population and Sub-Groups

Adult OHCA patients admitted to any of the six intensive care units (ICUs) in three tertiary hospitals in Stockholm, Sweden, and recorded in the Swedish register for cardiopulmonary resuscitation (SRCR) and the Swedeheart register between 2010–2016 were eligible for study inclusion. Patients receiving levosimendan >24 h from ICU admission were excluded.

A sub-group classification was further performed regarding initial rhythm (shockable versus non-shockable) and time to starting treatment with levosimendan from ICU admission (0–6 h, 6–12 h and 12–24 h).

### 2.3. Data Sources

#### 2.3.1. The Swedish Register for Cardiopulmonary Resuscitation (SRCR)

The SRCR includes all Emergency Medical Service (EMS) organisations in Sweden and includes the vast majority of OHCA cases in Sweden in whom cardiopulmonary resuscitation (CPR) was attempted [19]. It is a national quality register administered by the National Board of Health and Welfare and has been previously described in detail [20]. The EMSs report data in accordance with Utstein guidelines [21]. The register predominantly contains pre-hospital variables and data on 30-day survival [22].

#### 2.3.2. Centricity Critical Care (CCC)

Centricity Critical Care CCC (GE Healthcare) is a patient data system used in ICUs in Stockholm to collate vital parameters, blood gas analysis, administered medicines and fluids, and data from ventilators and other apparatus. The system is linked with a patient note system which contains administrative data and laboratory results. The collected information includes diagnosis and intervention codes and ICU-outcome.

#### 2.3.3. The SWEDEHEART Register

The SWEDEHEART register components include SCAAR (Swedish Coronary Angiography and Angioplasty Register) and RIKS-HIA (Register of Information and Knowledge about Swedish Heart Intensive Care Admission). SCAAR contains information on patients from hospitals performing Coronary angiography and percutaneous coronary intervention in Sweden. Coronary angiographic data are collected in a predefined manner according to data registration standards for clinical practice.

### 2.4. Exposure and Outcome

The independent exposure variable was levosimendan treatment initiated within 24 h from ICU admission in OHCA patients. The outcome measure was 30-day mortality.

### 2.5. Statistical Analysis

Data are presented as the total number of patients and proportion (%) in each group. Differences between groups, including outcome, were assessed by using the Chi-square test for categorical data and Wilcoxon rank sum test for continuous data. The results were regarded as significant if these tests yielded a *p*-value of equal to or less than 0.05. Logistic regression was used to determine the association between levosimendan treatment and 30-day mortality. The following variables were included in the multivariable analysis: age (years), gender, bystander-CPR (yes/no), witnessed event (yes/no), acute myocardial infarction (yes/no) and initial rhythm (shockable versus non-shockable rhythm). The associations are presented as an odds ratio (OR) with 95% confident interval (CI). In addition, to assess outcome propensity, score matching was conducted using 1:1 matching (age, sex, witnessed arrest, shockable rhythm, myocardial infarction, EMS response time and bystander CPR rate) and nearest neighbor with a caliper width of 0,1 to assess outcome. All analyses were performed using Stata SE Stata SE (14.2 StataCorp LLC, College Station, TX, USA).

## 3. Results

Among 1015 OHCA admitted to the ICU between 10 January 2010 and 31 December 2016, a total of 940 patients were included in the study. Overall, 94 (10%) patients received levosimendan <24 h from admission, and 846 (90%) did not receive any levosimendan (Figure 1).

### 3.1. Patient Characteristics and Treatment in All Patients

Baseline demographic and resuscitation characteristics are shown in Table 1. In the levosimendan group there were more men compared to the group not treated with levosimendan (81% vs. 67%, *p* = 0.007) (Table 1). There were no differences between the groups regarding age. The proportions of patients with shockable rhythm (66% vs. 37%, *p* < 0.001), acute myocardial infarction (AMI) (47% vs. 24%, *p* < 0.001) and circulatory shock with the need for vasoactive support, e.g., Noradrenaline (98% vs. 61%, *p* < 0.001) were higher in the levosimendan group (Table 1 and Table 2). The proportion of ST-elevation myocardial infarction (STEMI) among those with MI was higher in the levosimendan group (75% vs. 52%, *p* = 0.005) and percutaneous coronary intervention was higher in the levosimendan group (39% vs. 13%, *p* < 0.001) (Table 1). The median time to the initiation of levosimendan treatment was 6.1 h (IQR 2.4–10.1 h) with a median dose of 11.66 mg (IQR 10.66–13.40). The median length of stay in ICU was longer for the levosimendan group, with four vs. two days (IQR 2–7 vs. 1–4, *p* < 0.001) (Table 2).

### 3.2. Outcome in All Patients

The 30-day mortality rate was 57% (*n* = 54) in the group treated with levosimendan versus 70% (*n* = 590) in the group not treated with levosimendan (Table 2). The unadjusted OR for 30-day mortality was 0.59 (95% CI, 0.38–0.90, *p* = 0.02). When adjusted for age (year), sex, bystander-CPR (yes/no), witnessed event (yes/no), myocardial infarction (yes/no) and initial rhythm (shockable/non-shockable), the OR for 30-day mortality in the levosimendan group was 0.94 (95% CI, 0.56–1.57, *p* = 0.82) (Table 3 and Appendix A).

In the propensity score analysis, we studied patients with circulatory shock defined as patients needing noradrenaline. In this cohort, there was no association between treatment with Levosimendan and 30-day mortality, with 60% of patients in the control group requiring noradrenaline versus 59% of the patients treated with Levosimendan, *p* = 1.0 (Appendix A).

### 3.3. Patient Characteristics and Treatment in Patients with Initial Shockable Rhythm

In the subgroup of patients with initial shockable rhythm (*n* = 373), 17% (*n* = 62) received levosimendan < 24 h and 83% (*n* = 311) did not (Table 1). There were no differences in baseline characteristics between the groups (Table 1). The proportion with MI (61% vs. 42%, *p* < 0.007) and the need for vasoactive support (100% vs. 71%, *p* < 0.001) were higher in the levosimendan group (Table 2). The proportion of STEMI among those with MI did not differ between the groups (71% vs. 61%, *p* = 0.27) (Table 1). The median time to the initiation of levosimendan treatment was 3.9 h (IQR 1.9–7.9). The ICU length of stay was longer for the levosimendan group, with a median time of five vs. three days (IQR 3–6 vs. 1–6, *p* < 0.015) (Table 2).

### 3.4. Outcome in Patients with Initial Shockable Rhythm

The 30-day mortality rate in this subgroup population of patients with initial shockable rhythm, was 50% (*n* = 31) in the levosimendan group vs. 41% (*n* = 127) in the group not treated with levosimendan (Table 2). The unadjusted OR 30-day mortality in the levosimendan group was 1.45 (95% CI, 0.84–2.50, *p* = 0.18) and the adjusted OR for 30-day mortality in the levosimendan group was 1.35 (95% CI, 0.75–2.43, *p* = 0.32), (Table 3, Appendix A).

### 3.5. Exploratory Analyses

In an exploratory analysis of time to starting treatment with levosimendan, (0–6 h, 6–12 h and 12–24 h), the adjusted OR for 30-day mortality was 0.43 (95% CI, 0.21–0.89, *p* = 0.025) when levosimendan treatment was started 0–6 h from ICU admission compared to if treatment was started at a later stage (Table 3). Baseline demographic and resuscitation characteristics as well as ICU-interventions and outcome for these subgroups are shown in Appendix A.

## 4. Discussion

We conducted a registry-based observational study with the aim of describing the usage and effects of levosimendan within 24 h from ICU-admission in OHCA patients. One of our main findings was that only 10% of the OHCA population received inotropic support with levosimendan in the first 24 h after ICU admission. In the population treated with levosimendan, we see primarily men, patients with an initial shockable rhythm and acute myocardial infarction. Further, we observed an association with a higher need for vasopressor agents and additional inotropic support in the group of patients treated with levosimendan. The outcome analyses must be regarded as exploratory due to the low number of patients treated with levosimendan and the limitations in adjusting for disease severity (e.g., severity of cardiogenic shock) and multiorgan failure. In the adjusted outcome analysis, no differences in 30-day mortality between the groups were observed. Similar findings were seen in the propensity score matching analysis where we selected patients with circulatory shock requiring noradrenaline. Within the group who received levosimendan, we could identify that initiating treatment with levosimendan within the first 6 h after ICU-admission was associated with improved 30-day survival compared to when initiating treatment at a later stage. However, when interpreting these findings, it is important to acknowledge that a better matching on disease severity, indication for use and other confounders that cannot be identified in this register data needs to be performed to derive any conclusions for outcome from this dataset.

In this study, the patients receiving Levosimendan most often had an underlying cardiac etiology with initial shockable rhythm and a high proportion of acute myocardial ischemia. Several of the mechanisms of action of levosimendan may mitigate the ischemic-reperfusion injuries that may ensue after these cardiac arrests. There are multiple interacting processes that contribute to the reversible deterioration of cardiac function after a cardiac arrest, leading to acute cardiac dysfunction superimposed on underlying structural heart disease. Ischemic reperfusion, catecholamines and cytokines, all constitute a threat to normal cardiovascular function. In addition to levosimendan as the main indication as an inodilator in acute severe HF, the activation of mitochondrial K_ATP_-channels may be clinically advantageous after successful CPR. The activation of these channels potentially conveys a vital role in cardioprotection [23]. In addition, in cardiac surgery patients with acute kidney injury, levosimendan has been shown to improve renal blood flow via renal vasodilation, but with no or little effect on glomerular filtration rates [24]. The pre-and postconditioning properties of levosimendan have been investigated in experimental ischemia-reperfusion models indicating cardioprotective effects of the drug with a reduced infarction size and the recovery of myocardium at risk [25,26]. The effects seem to be linked to the combination of an improved hemodynamic performance and the activation of mitochondrial K_ATP_-channels in myocytes. In addition, these effects have been observed in other cell types as well (i.e., brain, liver and kidneys) [27,28]. Thus, the different mechanisms of action compared to other inotropic agents may have important clinical implications. In particular, in patients with acute HF or in patients with pre-existing severe ventricular impairment undergoing planned myocardial stress, the administration of levosimendan may potentially be used to bridge the patient through the critical phase [29].

In the present cohort, we observed that patients, after the successful resuscitation of mostly ischemic-ventricular fibrillation cardiac arrest, were more frequently treated with levosimendan. Since shockable rhythms are associated with an improved outcome compared to other CA rhythms, these patients may be a specific group for whom full measures are taken more frequently. Interestingly, for the patients treated with levosimendan, first-hand inotropic treatment options were more often conventionally used inotropic drugs (i.e., dobutamine, adrenaline), usually used in combination with one or more vasopressors with levosimendan being a later inotropic alternative. The vasodilating property of levosimendan, with the potential to aggravate an already existing hypotension in situations with compromised hemodynamic functions, may be one reason for the delayed or lack of levosimendan use observed in this study. However, the use of vasodilation in cardiac arrest has been investigated previously. Yannopoulos et al. investigated sodium nitroprusside intra-arrest in a porcine model of ischemic cardiac arrest. In their study, sodium nitroprusside was associated with improved ROSC as well as short term survival [30]. Recently, in a blinded randomized placebo-controlled study in swine investigating the effects of the early administration (intra-arrest) of levosimendan vs. placebo on survival in an ischemic cardiac arrest model, we demonstrated that there was an increased survival of the animals treated with levosimendan [13]. Interestingly, there was a less pressing need for both inotropic and vasopressor support in the animals treated with levosimendan despite a tendency towards more pronounced vasodilatation in the same group.

The literature is sparse regarding circulatory targets and the timing of vasoactive support after cardiac arrest. However, persistent vasoactive support requirements lasting more than 24 h after ROSC due to non-receding myocardial dysfunction and vasoplegia are associated with poor outcome [31]. Our exploratory finding of an association between early levosimendan initiation (<6 h from ICU admission) and improved survival warrants further investigation. One possible explanation might be a reduced need for catecholaminergic inotropic support since catecholamine in high doses has been described to worsen myocardial function [31]. In this context, with acute and severe HF subsequent to an OHCA, levosimendan, administrated at an early stage during decompensation, may improve hemodynamic and cardiac recovery, thus preparing a patient for the administration of beta blockers, mineralocorticoid receptor antagonists, SGLT2 inhibitors and angiotensin-converting enzyme inhibitor/angiotensin receptor-neprilysin inhibitors [32].

This is in line with several animal studies demonstrating an overall beneficial effect of the early administration of levosimendan in experimental cardiac arrest. In these studies, levosimendan treatment was initiated either intra-arrest or near the arrest time [14,15,16,17]. These studies indicate promising hemodynamic and organ-protective effects intra-arrest as well as in the following post-resuscitation period. However, in the clinical setting, there are only a few case reports where the intra-arrest administration of levosimendan in refractory cardiac arrest has been described [33,34,35,36]. It is also worth mentioning that only in the last year, several experimental studies examining the effects of levosimendan after cardiac arrest or in I/R-arrest have been conducted, which may encourage future clinical trials on the topic.

There has been a debate regarding the potential pro-arrhythmic properties of levosimendan but without a clear resolution. In this study, the initiation of anti-arrhythmic treatment (i.e., amiodarone) did not differ in frequency before or after the initiation of levosimendan (data not shown).

Finally, we could not observe any differences in 30-day mortality between patients treated with levosimendan within 24 h after an OHCA compared to those who did not. Even though the present study may be limited by, as discussed previously, a high risk of “confounding by indication” for both groups, it may indicate that levosimendan treatment after OHCA could be an early alternative for hemodynamic support not associated with an increased risk of death, not even in situations of severe compromised hemodynamics. Further prospective studies are needed to determine the timing and effects of levosimendan on cardiac performance and the outcome in OHCA-patients.

## 5. Limitations

There are several limitations to this study. Missing data present a hazard for retrospective studies which implies that data may be scarce in variables that are important when matching groups, such as cardiogenic shock, multi organ failure and the severity of these two conditions. Comorbidities and demographic information were not included in this study, nor were data of neurological outcome or echocardiography assessment. The observational design precludes any causal association and might introduce the risk of selection bias and confounding by indication. Further, the different ICUs within this study may use different post-arrest management algorithms, which we have not included in the analysis since post-arrest algorithms are overlapping and hard to describe succinctly as a variable in regression analysis.

## 6. Conclusions

In this observational study of OHCA patients, levosimendan was used in a limited patient group, mostly in those with initial shockable rhythms, acute myocardial infarction and with a high need for vasopressors. In this limited patient cohort, levosimendan treatment was not associated with 30-day mortality. However, a better matching of patient factors and indications for use needs to be performed to derive conclusions on associations with outcome. Further prospective studies are needed to determine the effects of levosimendan on cardiac performance and outcome in OHCA patients.

## Figures and Tables

**Figure 1 jcm-11-02621-f001:**
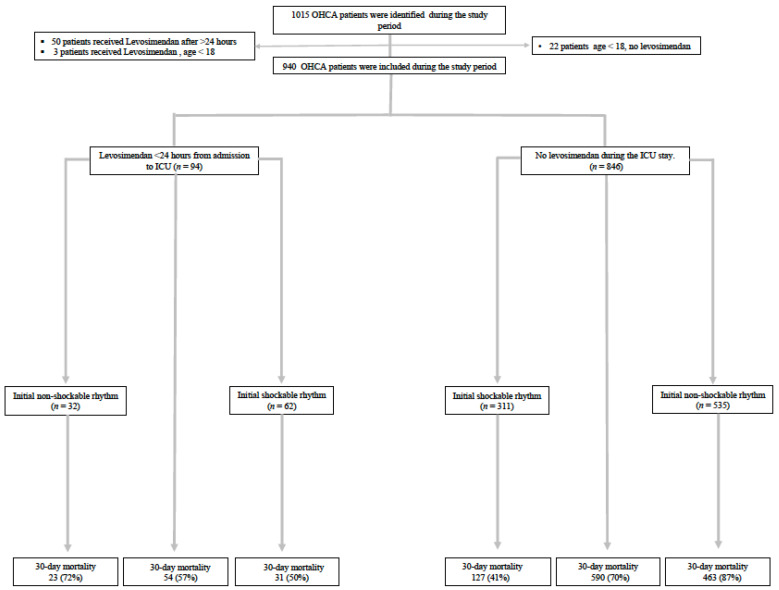
The CONSORT Flow Diagram of out-of hospital cardiac arrest patients admitted to the ICU in Stockholm, Sweden, 2010–2016. Abbreviations: OHCA = out-of-hospital-cardiac arrest; ICU = intensive care unit.

**Table 1 jcm-11-02621-t001:** Baseline characteristics in out-of-hospital-cardiac arrest patients admitted to the intensive care unit in Stockholm, Sweden, 2010–2016.

Characteristics	Levosimendan < 24 h (*n* = 94)	No Levosimendan (*n* = 846)	*p*-Value
**Sex, *n* = (%)**			
**Male**	76 (81)	568 (67)	0.007
**Age, year**			
Median, (IQR)	66 (60–74)	67 (56–77)	1.00
**Acute myocardial ischemia, *n* = (%)**			
Myocardial infarction, any	44 (47)	199 (24)	<0.001
STEMI	33 (75)	103 (52)	0.005
**CPR-characteristics, *n* = (%)**			
Shockable rhythm	62 (66)	311 (37)	<0.001
Witnessed event *	71/90 (79)	574/795 (72)	0.18
Bystander CPR *	53/90 (59)	364/792 (46)	0.02
Respons-time, minutes median (IQR) *	9 (7–14.5)	9 (7–13)	0.89
Coronary angiography with PCI	37 (39)	112 (13)	<0.001
**Patients with initial shockable rhythm**			
Characteristics	Levosimendan < 24 h (*n* = 62)	No Levosimendan (*n* = 311)	
**Sex, *n* = (%)**			
Male	50 (81)	247 (79)	0.83
**Age, year**	12 (19 )	64 (21)	
Median, (IQR)	66 (61–73)	66 (56–76)	0.68
**Acute myocardial ischemia, *n* = (%)**			
Acute myocardial infarction	38 (61)	132 (42)	0.007
STEMI	27 (71)	81 (61)	0.27
**CPR-characteristics**			
Witnessed event *	48/59 (81)	244/294 (83)	0.76
Bystander-CPR *	42/59 (71)	181/292 (62)	0.18
Respons-time, minutes median (IQR) *	10 (7–16)	9 (6–12)	0.24
Coronary angiography with PCI	32 (52)	90 (29)	0.001

Data are presented as median (IQR) for continuous measures, and *n* (%) for categorical measures. Abbreviations: STEMI = ST-elevation myocardial infarction; CPR = cardiopulmonary resuscitation; PCI = percutaneous coronary intervention. * missing values. In the variables with missing values we present nominator-denominator.

**Table 2 jcm-11-02621-t002:** Interventions and outcomes in out-of-hospital-cardiac arrest patients admitted to the intensive care unit in Stockholm, Sweden, 2010–2016.

Characteristics	Levosimendan < 24 h (*n* = 94)	No Levosimendan (*n* = 846)	*p*-Value
** *ICU-interventions and outcome* **			
*Vasoactive/inotropic support, n= (%)*			
Noradrenaline	92 (98)	515 (61)	<0.001
Adrenaline	14 (15)	55 (6)	0.003
Amiodarone	15 (16)	40 (5)	<0.001
Milrinone	14 (15)	6 (1)	<0.001
Dobutamine	18(19)	65 (8)	<0.001
Arginine vasopressin	1 (1)	8 (1)	0.91
*Cardiogenic shock **	13/58 (22)	52/326 (16)	0.23
*TTM **	45/71 (63)	237/629 (38)	<0.001
*ICU stay, days, median (IQR) **	4 (2–6)	2 (1–4)	<0.001
Mortality, 30-days, *n* = (%)	54 (57)	590 (70)	0.02
** *Patients with initial shockable rhythm* **			
Characteristics	Levosimendan < 24 h (*n* = 62)	No Levosimendan (*n* = 311)	
** *ICU-interventions and outcome* **			
Vasoactive/inptropic support, *n* = (%)			
Noradrenaline	62 (100)	222 (71)	<0.001
Adrenaline	8 (13)	10 (3)	0.001
Cordarone	9 (14)	25 (8)	0.11
Milrinone	9 (14)	4 (1)	<0.001
Dobutamine	15 (24)	24 (8)	<0.001
Arginine vasopressin	1 (1,7)	0	0.03
Cardiogenic shock *	11/48 (23)	29/208 (14)	0.12
TTM *	35/49 (71)	163/258 (63)	0.27
ICU-stay days, median (IQR)	5 (3–6)	3 (1–6)	0.02
Mortality, 30-days, *n* = (%)	31 (50)	184 (59)	0.18

Data are presented as median (IQR) for continuous measures, and *n* (%) for categorical measures. Abbreviations: ICU = intensive care unit; TTM = targeted temperature management. * missing values. In the variables with missing values we present nominator-denominator.

**Table 3 jcm-11-02621-t003:** Multivariable regression analyses showing the association with Levosimendan treatment and 30-day mortality in out-of-hospital-cardiac arrest patients admitted to ICU in Stockholm, Sweden 2010–2016. All analyses were adjusted for age, sex bystander-CPR, witnessed event, myocardial infarction and initial rhythm.

Variable	Odds Ratio (95% CI)	*p*-Value
Levosimendan < 24 h	0.94 (0.56–1.57)	0.82
Levosimendan < 24 h in patients with shockable rhythm	1.35 (0.75–2.43)	0.32
Levosimendan start 0–6 h	0.43 (0.21–0.90)	0.03
Levosimendan start 6–12 h	2.89 (1.14–7.32)	0.03
Levosimendan start 12–24 h	1.06 (0.34–3.32)	0.916

Abbreviations: OHCA; out-of-hospital-cardiac arrest, ICU = intensive care unit; CPR = cardiopulmonary resuscitation; CI = confidence interval.

## Data Availability

The study present data from The Swedish Register for Cardiopulmonary Resuscitation, The Centricity Critical Care patient data and monitoring system in the Stockholm region, and The SWEDEHEART register. The data are not publicly available in accordance with ethical approval and institutional regulations of patient data management.

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
