# Peer review of "The Use of Levosimendan after Out-of-Hospital Cardiac Arrest and Its Association with Outcome—An Observational Study"

_jcm, 2022, doi:10.3390/jcm11092621_

Round 1

Reviewer 1 Report

The scientific work was carried out with great rigor, presenting the methods and results clearly.
I appreciate the rigor used in the results without drawing conclusions that are not supported by the data. The conclusion of the authors who believe that an expansion of the patients treated is necessary in order to express an opinion seems to me to be the right final comment for this work.

Author Response

Point 1. The scientific work was carried out with great rigor, presenting the methods and results clearly. I appreciate the rigor used in the results without drawing conclusions that are not supported by the data. The conclusion of the authors who believe that an expansion of the patients treated is necessary in order to express an opinion seems to me to be the right final comment for this work.

Response: Thanks for assessing the manuscript and the encouraging comment. 

Reviewer 2 Report

Rysz et al reported an observational study regarding the relationship between the use of levosimendan after out of hospital cardiac arrest and 30 days mortality. The main results was that Levosimendan treatment was not associated with 30-day survival, representing a comparable alternative to inotropes. some aspects should be improved:

1) Please tables should be displaced near the text, after their citation 

2) please use more abbreviation such as HF for heart failure. Citations should be cited first time they appear in the text, abstract, tables. 

3) The study highlights the importance of early use of Levosimendan. Due to its systemic effects on myocardium, bloode vessels (thanks to the effect on K-ATP channels). Please add a further discussion regarding the role of Levosimendan on kidneys and eGFR (see Crit Care. 2021 Jun 12;25(1):207. doi: 10.1186/s13054-021-03628-z.). In this context Levosmendan, adiminstrated early during the acute phase of HF decompensation may improve hemodynamic, preparing the patient to adiministration of BB, MRAs, SGLT2i and ARNI (see J Clin Med. 2022 Feb 6;11(3):857. doi: 10.3390/jcm11030857). 

Author Response

Rysz et al reported an observational study regarding the relationship between the use of levosimendan after out of hospital cardiac arrest and 30 days mortality. The main results was that Levosimendan treatment was not associated with 30-day survival, representing a comparable alternative to inotropes. some aspects should be improved:

1) Please tables should be displaced near the text, after their citation 

Thanks for the comment. We have now inserted references to the tables after each statement in the results section.

2) please use more abbreviation such as HF for heart failure. Citations should be cited first time they appear in the text, abstract, tables. 

Point well taken. We have done the changes suggested and explained and used the abbreviation OHCA, HF, AMI, CPR, ICU and for some of the registries used.

3) The study highlights the importance of early use of Levosimendan. Due to its systemic effects on myocardium, bloode vessels (thanks to the effect on K-ATP channels). Please add a further discussion regarding the role of Levosimendan on kidneys and eGFR (see Crit Care. 2021 Jun 12;25(1):207. doi: 10.1186/s13054-021-03628-z.). In this context Levosmendan, adiminstrated early during the acute phase of HF decompensation may improve hemodynamic, preparing the patient to adiministration of BB, MRAs, SGLT2i and ARNI (see J Clin Med. 2022 Feb 6;11(3):857. doi: 10.3390/jcm11030857). 

Thanks for the very valuable comment. This is indeed very interesting. We have expanded the discussion somewhat to include the influence on renal blood flow and GFR. In addition, as suggested we have discussed the potential advantages of early administration of Levosimendan use in order prepare for other HF medications. The changes is highlighted in page 5 and 6 with tracked changes.

Reviewer 3 Report

The authors present registry data on levosimendane use in patients with OHCA.

While the topic is interesting the work fails to offer plausible findings.

The methods are not appropriate to investigate effects on outcome data in this way. Levosimendane group needs to be compared to a control group with cardiogenic shock. It makes no sense to compare these patients to "all comers with OHCA". These are completely different patient populations. A better matching needs to be performed to derive any conclusion from this dataset.

Author Response

The authors present registry data on levosimendane use in patients with OHCA.

While the topic is interesting the work fails to offer plausible findings.

The methods are not appropriate to investigate effects on outcome data in this way. Levosimendane group needs to be compared to a control group with cardiogenic shock. It makes no sense to compare these patients to "all comers with OHCA". These are completely different patient populations. A better matching needs to be performed to derive any conclusion from this dataset.

Thanks for this very important comment. This is indeed true that we may assume that Levosimenden was used on the indication of cardiogenic shock and therefor the comparison with other cardiac arrest populations may be skewed and unfair. However, from the data we have available from the SWEDEHEART registry and the Stockholm ICU registry the proportion of patients with cardiogenic shock is low (23 vs 14%) and we were not able to use that variable in the adjustments or matching. Probably this is partly due to errors in registration leading to missing data, but we can only speculate on this matter.

Instead to use cardiogenic shock for matching, we have updated the propensity score matching analysis, now including only patients in both groups with circulatory shock (e.g. with need of Noradrenaline as vasopressor) to get a better match in population. The supplementary table 2a, 2b has been updated. The last sentence in the conclusion (abstract and manuscript) has been rephrased to show the readers that this result is based on a limited study population. We have also expanded this limitation in the beginning of the discussion on page and in the section of limitations on page 6.
